# Fibrosis of Peritoneal Membrane, Molecular Indicators of Aging and Frailty Unveil Vulnerable Patients in Long-Term Peritoneal Dialysis

**DOI:** 10.3390/ijms24055020

**Published:** 2023-03-06

**Authors:** Patrícia Branco, Rita Calça, Ana Rita Martins, Catarina Mateus, Maria João Jervis, Daniel Gomes Pinto, Sofia Azeredo-Lopes, Antonio Ferreira De Melo Junior, Cátia Sousa, Ester Civantos, Sebastian Mas-Fontao, Augusta Gaspar, Sância Ramos, Judit Morello, Fernando Nolasco, Anabela Rodrigues, Sofia Azeredo Pereira

**Affiliations:** 1Nephrology Department, Hospital Santa Cruz, Centro Hospitalar de Lisboa Ocidental (CHLO), 2790-134 Lisboa, Portugal; mateus.meg@gmail.com (P.B.); arrcalca@gmail.com (R.C.); anarita.mateus@gmail.com (A.R.M.); ana.catarina.mateus@gmail.com (C.M.); magaspar@chlo.min-saude.pt (A.G.); 2iNOVA4Health, NOVA Medical School|Faculdade de Ciências Médicas, NMS|FCM, Universidade Nova de Lisboa, 1150-082 Lisboa, Portugal; melo.junior@nms.unl.pt (A.F.D.M.J.); catia.moreirasousa@nms.unl.pt (C.S.); judit.morello@nms.unl.pt (J.M.); febnolasco@gmail.com (F.N.); 3Centro Clínico Académico de Lisboa, 1159-056 Lisboa, Portugal; 4Surgery Department, Hospital Santa Cruz, Centro Hospitalar de Lisboa Ocidental (CHLO), 2740-134 Lisboa, Portugal; mjfernandes@chlo.min-saude.pt; 5Pathology Department, Hospital Santa Cruz, Centro Hospitalar de Lisboa Ocidental (CHLO), 2740-134 Lisboa, Portugal; danielgomespinto@gmail.com (D.G.P.); sanciaramos@hotmail.com (S.R.); 6CHRC, NMS|FCM, Universidade Nova de Lisboa, 1150-082 Lisboa, Portugal; sofia.azeredo@nms.unl.pt; 7Department of Statistics and Operational Research, Faculdade de Ciências, Universidade de Lisboa, 1749-016 Lisboa, Portugal; 8Renal, Vascular and Diabetes Research Laboratory, IIS-Fundación Jiménez Díaz, Universidad Autónoma de Madrid, Spanish Biomedical Research Centre in Diabetes and Associated Metabolic Disorders (CIBERDEM), 28029 Madrid, Spain; ecivamar@uax.es (E.C.); smas@quironsalud.es (S.M.-F.); 9UMIB—Unidade Multidisciplinar de Investigação Biomédica, ITR—Laboratory for Integrative and Translational Research in Population Health, 4050-313 Porto, Portugal; rodrigues.anabela2016@gmail.com; 10Departamento de Nefrologia, ICBAS—Instituto de Ciências Biomédicas Abel Salazar, Universidade do Porto, Centro Hospitalar Universitário do Porto (CHUdsA), 4050-345 Porto, Portugal

**Keywords:** α-Klotho, galectin-3, uremic toxins, cardiovascular toxicity, chronic kidney disease

## Abstract

Peritoneal membrane status, clinical data and aging-related molecules were investigated as predictors of long-term peritoneal dialysis (PD) outcomes. A 5-year prospective study was conducted with the following endpoints: (a) PD failure and time until PD failure, (b) major cardiovascular event (MACE) and time until MACE. A total of 58 incident patients with peritoneal biopsy at study baseline were included. Peritoneal membrane histomorphology and aging-related indicators were assessed before the start of PD and investigated as predictors of study endpoints. Fibrosis of the peritoneal membrane was associated with MACE occurrence and earlier MACE, but not with the patient or membrane survival. Serum α-Klotho bellow 742 pg/mL was related to the submesothelial thickness of the peritoneal membrane. This cutoff stratified the patients according to the risk of MACE and time until MACE. Uremic levels of galectin-3 were associated with PD failure and time until PD failure. This work unveils peritoneal membrane fibrosis as a window to the vulnerability of the cardiovascular system, whose mechanisms and links to biological aging need to be better investigated. Galectin-3 and α-Klotho are putative tools to tailor patient management in this home-based renal replacement therapy.

## 1. Introduction

Peritoneal dialysis (PD) is a home-based modality of renal replacement therapy and a good option for patients with chronic kidney disease (CKD). Independently of the chronological age, this population commonly presents an accelerated aging process affecting skeletal, immune, renal, and cardiovascular systems [1]. Therefore, the risk of mortality in CKD patients is increased by 10- to 20-fold in comparison to individuals with normal renal function [2]. Moreover, cardiovascular toxicity caused by uremia represents a major factor for the increased mortality in dialysis programs [3].

The dialytic capacity of the peritoneal membrane is pivotal in PD, but the integrity of the membrane in the uremic patient might have been overlooked. The existence of a high person-to-person variability in the status of the membrane before the start of PD was recently reported and related to the anti-aging molecule α-Klotho [4].

Deficiency of α-Klotho is well known to be involved in damage of the cardiovascular system, atherosclerosis, skin atrophy and osteoporosis, traits commonly associated with human aging [5,6]. Those traits also overlap with the manifestations of CKD [7], suggesting that α-Klotho might be an important player in PD outcomes. α-Klotho is changed in uremia [8], which is recognized as a disbalance between protective and harmful molecules [9]. In uremia, the concentrations of proteins associated with mechanisms underlying early aging can be affected, such as those related to inflammation and fibrosis in multiple organs/tissues [10].

In fact, α-Klotho was shown to be a uremic molecule implicated in the vulnerability of the peritoneal membrane, expressed as submesothelial fibrosis [4]. As more vulnerable peritoneal membranes were associated with low circulating α-Klotho, we herein hypothesized that α-Klotho might represent a multifaceted marker of both the survival of the membrane and the survival of the patient. Therefore, we conducted a prospective longitudinal observational study in a cohort of incident PD patients to investigate the impact of uremic toxins related to aging, peritoneal membrane status and the patient’s frailty in long-term PD outcomes.

## 2. Results

### 2.1. Baseline Characterization of Study Population

This observational prospective cohort study included 58 patients, followed for 60 months. A total of 31% were female. At baseline, patients were 56 (30–79) years old with a median renal residual function assessed by rGFR of 7 (4–10) mL/min/1.73 m^2^. The underlying renal diseases were diabetic renal disease (20%), chronic glomerulonephritis (20%), hypertensive nephrosclerosis (23%), autosomal dominant polycystic kidney disease (11%) and chronic pyelonephritis (10%). Twenty-two patients had fibrosis of the peritoneal membrane at the baseline of the study.

Concerning dialysis parameters, 2 and 32 patients were fast and average-fast transporters, respectively, and 94% of patients had good efficacy of dialysis.

Regarding therapeutics, patients with atherosclerosis artery diseases (40%) were treated with the highest tolerated dose of statins and antiplatelet therapy. In addition, all patients were on inhibitors of renin–angiotensin axis (IECA or ARA) and diuretic therapy. Eighteen patients (31%) were on spironolactone, which was mainly added in those with fibrosis of the peritoneal membrane before the start of PD. A total of 12 patients were on beta-blockers and 30% were on other antihypertensive drugs. The number of patients in treatment for mineral bone disease was low.

The normalized protein catabolic rate (nPCR) was 0.99 (0.79–1.09) g/Kg/day and 47% had proper nutrition. A total of 10 patients were vulnerable and 5 were frail according to the Edmonton scale.

The baseline variables of the study were analyzed according to the biopsy score of the membrane (Table 1), which considers submesothelial compact zone thickness (STM), vasculopathy and inflammation [4]: Score 0 represents no fibrosis, no vasculopathy, nor inflammation; Score 1: no fibrosis, but vasculopathy and/or inflammation; Score 2: fibrosis with/without vasculopathy and/or inflammatory changes.

Overall, at baseline, patients with membrane fibrosis received more spironolactone, antiplatelet and statins therapy. In the S2 group (fibrosis), more than half of the patients had peripheral arterial disease (PAD). While the biopsy score was not related to the age of the patients, the cutoff for the level of circulating α-Klotho (anti-aging molecule) that discriminated the existence of peritoneal membrane fibrosis before the start of PD was defined by performing a ROC curve (AUC = 0.860, *p* = 4 × 10^−6^). This cutoff was established at 742 pg/mL, with 83% sensitivity (to detect fibrosis) and 71% specificity (to detect no fibrosis).

### 2.2. Impact of the Status of the Peritoneal Membrane and Age-Related Indicators in PD-Related Outcomes

Regarding the long-term outcomes of the study, the minimum time on PD was 13 months and the median time was 42 (30–58) months. Technical failure during the follow-up period occurred in 41% of patients, with a median time until failure of 40 (26–56) months.

A total of 27 patients (47%) had a MACE during the study, with a minimum time for MACE of 8 months and a median time of 17 (12–31) months.

Next, we investigated the relation of study outcomes with the status of the membrane (biopsy score, STM, α-Klotho levels with a 742 pg/mL cutoff as a surrogate of fibrosis) and the age-related baseline indicators (age, serum biomarkers, frailty).

### 2.3. Status of Peritoneal Membrane, Age-Related Indicators and Technical Failure of Peritoneal Dialysis

Contrary to our initial hypothesis, the status of the membrane was not associated with technical failure (Table 2). Overall, the patients with PD failure, compared to those without, were older, had higher frailty scores, were more likely to be on calcium channel blockers (Table 2) and presented higher circulating galectin-3 at the study baseline. The use of icodextrin solutions, glucose applied, or diabetes were not associated with failure (Table 2) or time to PD failure (Table 3).

In addition, the galectin-3 was also related to the time until PD failure (Table 3). A cut-off of galectin-3 to discriminate PD failure was established at 8.88 ng/mL (sensitivity = 92% and specificity of 46%), which was also associated with the survival of the peritoneal membrane (Figure 1A). This cut-off was independently associated with PD failure in an adjusted model to age, PAD, and calcium channel blockers (CCB) (Figure 1B), wherein age, frailty score and icodextrin did not account for the prediction of time to PD failure.

### 2.4. Peritoneal Membrane, Age-Related Indicators and Major Cardiovascular Event

While not related to PD failure, the presence of membrane fibrosis at the study baseline was associated with occurrence of MACE (Table 2) and time to MACE (Table 3, Figure 2A). Both endpoints were also related to age, frailty score, arterial atherosclerotic disease, use of statins, nPCR, beta-blockers and Kt/v (Table 2 and Table 3).

The existence of fibrosis in the peritoneal membrane at the study baseline was independently associated with time to MACE in an adjusted model to age, nutritional status and PAD (Figure 2B), wherein the frailty score or heart failure did not account for the prediction of the time to PD failure. This multivariate association was maintained when the membrane status was inferred by the non-invasive surrogate α-Klotho, using the identified cut-off for α-Klotho of 742 pg/mL instead of biopsy score (Figure 2C). The association of time until the occurrence of MACE with atherosclerotic disease might be inferred by the use of antiplatelet therapy (Figure 2D), maintaining α-Klotho cutoff as an independent factor in the model, together with age and antiplatelet use at the study baseline. The estimated survival probability for time to MACE discriminated by α-Klotho levels in an adjusted model to age, frailty, nPCR, rGFR and use of antiplatelet drugs is represented in Figure 2E.

Overall, our results suggest a link between the vulnerability of a patient’s cardiovascular system and the status of the peritoneal membrane. In addition to age, lower α-Klotho and PAD were also predictors of cardiovascular risk over time in different multivariate models.

### 2.5. Peritoneal Membrane, Age-Related Indicators and All-Causes Mortality

A total of six deaths occurred during the study, five related to cardiovascular disease and 1 to malignancy, which were not related to the biopsy score of the peritoneal membrane. Cardiovascular mortality and survivor groups had similar age, scores of biopsies and frailty as well as similar levels of serum biomarkers.

## 3. Discussion

Our data provides new information about the links between the peritoneal membrane, uremia and PD outcomes. We found that blood levels of galectin-3 represent a putative tool to identify patients at higher risk of PD failure. In addition, and contrary to our initial hypothesis, the baseline membrane fibrosis was not a predictor of technical failure, time to failure or all-causes mortality in PD. Instead, the status of the peritoneal membrane was related to MACE and time until occurrence of MACE, which can be inferred by circulating α-Klotho.

The rationale for the choice of the pre-PD molecules was driven by the hypothesis that prematurely aged phenotypes of the peritoneal membrane could be associated with poorer long-term PD outcomes. These phenotypes are difficult to predict only from demographic characteristics, but could be favored by a uremic toxic environment, patients’ frailty and aging. Therefore, a group of aging-related indicators was investigated as predictors of PD outcomes. PD outcomes were associated with uremic molecules, but not with the frailty test applied. This test was chosen due to its simplicity and ease for daily clinical practice and validation in Portuguese [11,12].

The person-to-person variability in membrane status and functions, even before the start of PD [4], is likely to be driven by genetic and non-genetic factors [13,14,15,16]. The latter includes exposure to glucose, peritonitis, loss of residual renal function, inflammation and uremia [4,17,18]. In this context, better knowledge about aging-related uremic molecules might fulfill clinicians’ aims for accessible risk stratification tools for tailored prescriptions. Foreseeing a proof of concept that uremia-related mechanisms impact both membrane and patient survival, we selected a panel of proteins reported to be associated with aging, inflammation and fibrosis in other organs/tissues [19,20,21,22,23,24,25,26,27,28,29,30,31,32,33].

We found that the status of the membrane (evaluated by histomorphology, STM and by a surrogate α-Klotho cutoff) was not associated with changes in the functions of the peritoneal transport. Moreover, the pre-PD membrane status was not predictive of long-term survival of both peritoneal membrane and patients.

As α-Klotho is associated with fibrosis of the peritoneal membrane [4], the absence of association between α-Klotho and PD failure was an unexpected finding. While it did not consider fibrosis, a previous study about the arteriolar structure concluded that membrane arteriolar frailty in CKD stage 5 patients follows with cardiovascular system damage [34]. Therefore, as α-Klotho is associated with arteriosclerosis and aging, our results might suggest that the peritoneal biopsy score reflects a vascular vulnerability more than the integrity of the membrane. This novel and overlooked dimension might account for the shared mechanisms of persistent uremic phenotype, premature aging, and fibrosis of different tissues. In fact, the membrane might not represent a risk factor but a marker of a particular cardiovascular vulnerability profile.

Substantial cardiovascular risk persists in CKD patients, despite the treatment of established cardiovascular risk factors such as arterial hypertension and dyslipidemia. Knowledge about the uremia profiles that might be predictors of these risks will pave the way for personalized interventions. Moreover, this knowledge aligns with the need for novel drugs to control the unbalanced status of protective and deleterious molecules that constitutes uremia.

Our data might support that even older, frail, at higher cardiovascular risk and/or with a worsened status of peritoneal membrane patients might take advantage of this home-based modality of renal replacement therapy because we did not find any association between frailty or peritoneal membrane status and mortality or survival in the technique. Attention must be paid to the combination of atherosclerotic arterial disease, namely PAD and low α-Klotho levels. α-Klotho is an anti-aging molecule that exerts beneficial effects on the endothelium [35]. Moreover, α-Klotho-deficient mice show increased vascular calcification [36,37], further supporting a beneficial cardiovascular role of α-Klotho and putative relevance of recombinant α-Klotho to control the burden of comorbidities in PD patients.

Differently from α-Klotho, there was a clear association with galectin-3 and PD failure. Galectin-3, which is secreted by macrophages, has been associated with an inflammatory and fibrotic phenotype [19,25,26,27,28,29]. Moreover, Béllon et al. (2011) showed that alternative activated macrophages or M2 phenotypes were present in the peritoneal effluent drained from patients, and were able to stimulate fibroblast proliferation and the loss of peritoneal function [30].

α-Klotho and galectin-3 share common characteristics, e.g., both are uremic toxins and have been related to fibrosis and inflammation. However, unlike α-Klotho, galectin-3 was associated with PD failure. Therefore, the differences found in PD outcomes between poor α-Klotho versus rich galectin-3 uremic profiles suggest different underlying mechanisms. Moreover, while low baseline α-Klotho was highly associated with cardiovascular disease, such an association was not found for galectin-3 (Table 2). Instead, our data indicated galectin-3 as a predictor of earlier PD failure. Further studies are necessary to validate this data, but a putative explanation for the galectin-3 result is that this molecule is a high-affinity binding protein for advanced glycation products [38] whose relation to poor membrane efficiency and survival is well accepted [39,40]. Of note, inhibitors of galectin-3 are currently investigated in clinical trials [41,42,43], although in areas other than PD.

Our study has several strengths. All biomarker measurements were performed in the same laboratory to ensure measurement consistency across the pooled cohort, and we analyzed an anatomical territory with fibrosis and achieved a long follow-up period.

However, the study has some limitations. Firstly, the strict inclusion criteria from a single PD center implied that a rather small sample was studied; serum biomarkers were only measured at baseline, which might have hampered finding associations with time-dependent outcomes (PD and MACE). Secondly, other parameters of adequacy such as nutrition and volemia were neglected in our research, which may have influence data analysis and affect our prediction of long-term outcomes of patients. Moreover, our data only focused on clinical examinations and basic personal information, not including environmental conditions such as psychosocial and economic dimensions, which can affect their clinical outcomes.

Further research might focus on the putative role of galectin-3 and α-Klotho as tools to tailor patient management in this home-based renal replacement therapy.

## 4. Materials and Methods

### 4.1. Study Design and Participants

This was a single center, prospective study with 60 months of follow-up that included incident patients at the PD Unit of Santa Cruz Hospital, Centro Hospitalar de Lisboa Ocidental, Portugal. The study was approved by the Ethics Committee of the NOVA Medical school, Faculdade de Ciências Médicas, NOVA University of Lisbon (Approval number 50/2019). The study was conducted according to the Declaration of Helsinki and Good Clinical Practices and complied with the European Union GDPR Legislation.

At the enrollment, patients were referred from the Nephrology consultation inside or outside the hospital for an information consultation. Patients were enrolled in a consecutive manner. The main purposes of the consultation were to assess the eligibility criteria for the renal substitution technique and to provide information to allow an informed choice. This consultation included a multidisciplinary team composed of a doctor, nurse, nutritionist, and social worker. Inclusion criteria were being over 18 years of age and having a stable clinical condition, defined by the absence of serious abdominal infections (diverticulitis, pancreatitis and cholecystitis) or active neoplasia, and on PD with biopsy of the peritoneal membrane. The exclusion criteria were having had previous aggressions to the peritoneal membrane (such as surgeries or peritonitis).

Non-autonomous patients were included for assisted PD whenever there was a caretaker. All patients signed informed consent.

### 4.2. PD Prescription

PD was started within 30 (21–44) days after the implantation of the catheter. All patients started on continuous ambulatory peritoneal dialysis (CAPD) and after the first year, 30% of patients switched to automated peritoneal dialysis (APD) and remained stable over the observation period. All patients were treated with dialysis solutions with a reduced content of glucose degradation products and a normal pH (Baxter^®^, Deerfield, MA, USA and Fresenius^®^, Bad Homburg, Germany). At baseline, no hypertonic solutions were used and a total of 38% of the patients received polyglucose. The major reasons to include polyglucose in the prescription was hydration status (41%), diabetes and the presence of basal peritoneal membrane fibrosis (30%). Prescriptions of amino-acid-containing solutions for PD were exclusive for diabetic patients. The daily quantity of administered glucose was maintained for CAPD, but increased in APD patients over time to achieve adequate ultrafiltration and fluid balance.

### 4.3. Baseline Variables

The following variables were assessed at study baseline

*(a)* Anthropometric, clinical and therapeutic variables*(b)* Pre-PD histomorphology score (Biopsy score)*(c)* Dialysis Efficacy. The dialysis efficacy was defined by the calculation of the urea clearance index (Kt/Vurea). This index considers the concentration of urea in blood, in the urine (renal clearance) and dialysate (peritoneal clearance), and the body surface. This was obtained from the collection of the PD effluent and of the patient’s urine for 24 h, prior to the scheduled visit at the PD unit. Weekly Kt/V values were calculated according to the recommendations of the kidney disease outcomes quality initiative [11,44] (DOQI). According to K/DOQI guidelines, the cut-off value of Kt/V ≥ 1.7 was set to define Dialysis Efficacy [45].*(d)* Profile of Peritoneal Transport. The category of Peritoneal Membrane Transport was determined using the peritoneal equilibration test (PET). The PET and dialysis efficacy were evaluated at the same time. In the night before the PET test, at home, a long overnight dwell of PD was performed using a 1.36% glucose dialysis fluid (isotonic). The next morning at the hospital, the PETs were performed using a 2 L dialysis solution with 3.86% or 4.25% glucose (hypertonic). The dialysate/plasma (D/P) creatinine ratio was measured and used to identify the patients as low (D/P creatinine 0.34–0.50, L), low average (D/P creatinine 0.50 to 0.65, LA), high average (D/P creatinine 0.65 to 0.80, HA) or high (D/P creatinine 0.81–1.03, H) transporters, according to previous definitions [46].*(e)* Fluid removal by the peritoneal membrane. The permeability of the membrane to fluid is defined by the ultrafiltration test, which compares the amount of drained dialysate with the 2 L of dialysis fluid instilled at start of the test. Ultrafiltration failure is defined as failure when the target of at least 400 mL of net ultrafiltration during a 4-h period of PD using 3.86% or 4.25% glucose solutions is not achieved (in absence of catheter malposition or mechanical dysfunction).*(f)* Residual renal function (RRF). The RRF was obtained through the creatinine clearance, which was calculated by collecting 24 h of urine before blood sampling and using conventional formulas and correcting the result for a body surface area of 1.73 m^2^/Kg.*(g)* Daily protein intake. Nutritional status is an important adequacy parameter in patients on dialysis. The normalized protein catabolic rate (nPCR) was calculated from the urea eliminated in urine and in dialysate and normalized to body weight. The recommended standard value of this parameter is ≥1 g/Kg/day.*(h)* Effluent CA125 levels. Effluent levels were measured using electrochemiluminescence (Elecsys, Roche diagnostics).*(i)* Levels of serum biomarkers. A panel of proteins related to aging and fibrosis was quantified at study baseline by Enzyme-Linked Immunosorbent Assay (ELISA), i.e., before the start of PD, which consisted of α-Klotho (Bionava assay), galectin-3 and FGF21 (Fibroblast growth factor 21) (Quantikine ELISAS, R&D systems), FGF23 (Fibroblast growth factor 23 c-terminal, Immunotopics), Tweak (Tumor necrosis factor-like weak inducer of apoptosis, Preprotech), TNFα (Tumor necrosis factor alfa, Preprotech) and hr-CRP (ultra-sensitive C-reactive protein assay using a Cobas c702 analyzer, Roche Diagnostics).*(j)* Frailty assessment. The Edmonton Frail Scale (FS) was selected as a simple assessment tool comprising eleven items focusing on different frailty dimensions [47].

Peritoneal and renal Kt/V urea and creatinine clearances, glomerular filtration rate (GFR), body surface area (BSA), and protein catabolic rate were calculated using Patient onLine (POL) software version 6.3 (Fresenius^®^, Bad Homburg, Germany). These variables were investigated as factors with impact on PD outcomes. All patients were followed up until death, PD drop-out, or 30 June 2019.

### 4.4. Study Outcomes

The primary outcomes were PD-related outcomes:-PD technique failure refers to ultrafiltration failure, peritonitis, or dialysis inefficacy. Patients were considered with no technical failure when achieving 60 months of follow-up.-Time for technique failure is the time on PD of each patient in the study until technical failure. Participants dropping PD out for reasons other than technical failure (switching to hemodialysis by option, kidney transplantation, transference to other PD centers or loss to follow-up) were censored.

The secondary outcomes were Cardiovascular Outcomes:*(a)* All-cause mortality*(b)* Major Cardiovascular event-Major Cardiovascular event (MACE) after 3 months on PD. MACEs were defined according to validated clinical criteria and included coronary heart disease (CHD), congestive heart failure (HF), acute myocardial infarction (AMI), acute cerebral infarction (ACI) and cardiac death caused by AMI, arrhythmias or HF. CHD was defined as ≥50% diameter stenosis of coronary arteries by either coronary angiography or CT angiography [48]. HF was diagnosed according to ESC guidelines for the diagnosis and treatment of chronic heart failure [49]. AMI was diagnosed according to ESC guidelines for the management of acute coronary syndromes [50]. ACI was defined as an acute neurological event lasting more than 24 h associated with clinical evidence of ischemic focus of the brain [51]. Cardiac death was defined as death caused by AMI, arrhythmias or CHF.-Time for MACE was defined for each patient as the time in the study until a MACE. Censored data were defined for those dropping out of the study without MACE or those achieving the end of the study without MACE.

### 4.5. Statistical Analyses

Categorical variables are presented as absolute (n) and relative frequencies (%); continuous non-normally distributed data are expressed as median (interquartile range). The Kruskal–Wallis test was used to assess differences between three or more independent groups. The Mann–Whitney U-test was used to assess differences between two independent groups. Potential associations between categorical data were analyzed using the Chi-Squared test. ROC curves were also used to identify cut-offs for potential blood biomarkers. Multiple Cox proportional hazards regression models were performed to assess potential predictors of survival, technique survival and the time to the occurrence of a cardiovascular event. The proportional hazards assumption was assessed through the Schoenfeld residual plots. All models were fit using the ‘survival’ R package [52,53].

The optimal cutpoints were obtained through the maximally selected rank statistics method (see [54] for more details) using the ‘*maxstat*’ R package [55] and considering the time until PD failure with the censor variable, indicating whether the patient suffered a technique failure or not.

Survival curves were generated using the Kaplan–Meier technique and tested using the log-rank test.

A confidence level alpha of 0.05 was considered throughout the study. Statistical analyses were performed with the R software, R version 4.2.0 and SPSS.

## Figures and Tables

**Figure 1 ijms-24-05020-f001:**
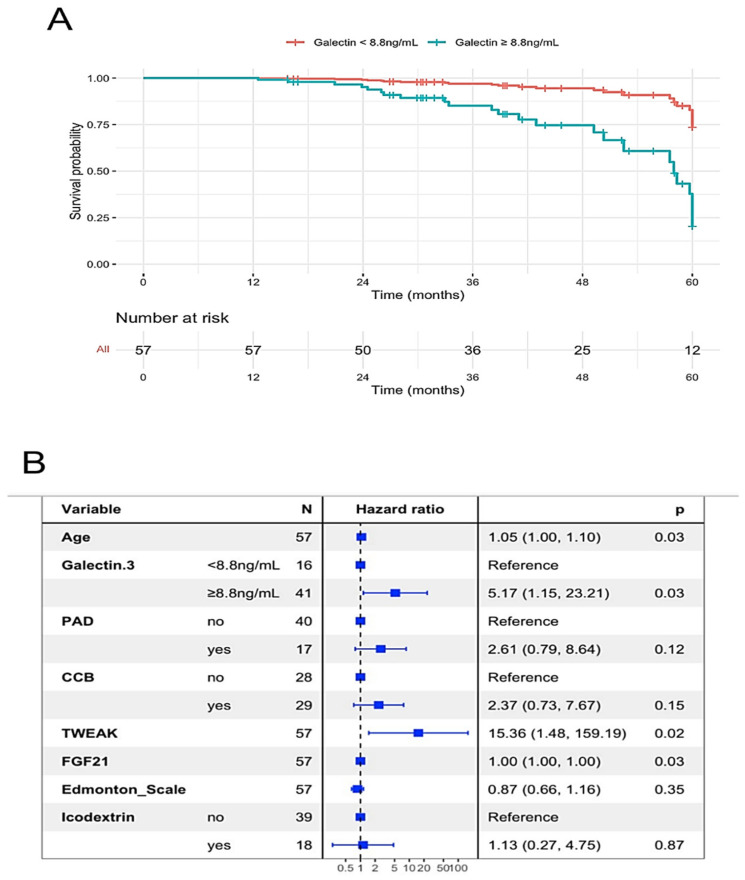
Predictors of time until failure in peritoneal dialysis (PD). (**A**) Survival curve of the peritoneal membrane according to galectin-3 cutoff of 8.8 ng/mL. (**B**) Multivariate Cox regression model of peritoneal dialysis failure using galectin-3 cutoff of 8.8 ng/mL. PAD—peripheral arterial disease; CCB—calcium channel blocker; TWEAK—Tumor necrosis factor-like weak inducer of apoptosis; FGF21—Fibroblast growth factor 21.

**Figure 2 ijms-24-05020-f002:**
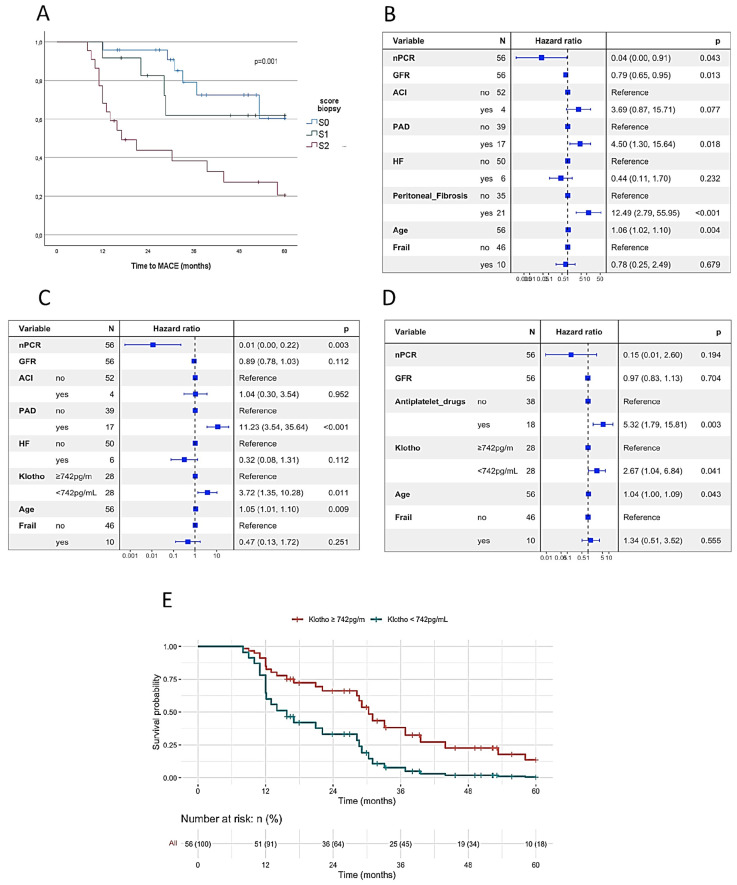
Predictors of time to major cardiovascular event (MACE). (**A**) Survival curve for time until MACE accordingly to peritoneal membrane status. (**B**) Multivariate Cox model that includes peritoneal fibrosis as an indicator of the membrane status adjusted for age, nutritional status, frailty and PAD. (**C**) Multivariate Cox model similar to B, but using α-Klotho cutoff as a surrogate of membrane fibrosis. (**D**) Multivariate Cox model using levels of circulating α-Klotho as surrogate of membrane fibrosis and using the antiplatelet therapy as an indicator of atherosclerotic arterial disease. (**E**) Survival curves for time until MACE according to α-Klotho cutoff of 742 pg/mL. GFR—glomerular filtration; PAD—peripheral arterial disease; CHD—coronary heart disease; HF—heart failure; nPCR—normalized protein catabolic rate.

**Table 1 ijms-24-05020-t001:** Baseline variables of the study according to the biopsy score of the peritoneal membrane.

Study Variable		Biopsy Score			
		S0 (n = 24)	S1 (n = 12)	S2 (n = 22)	*p*
Anthropometric & Clinical data	Women, n (%)	10 (42)	4 (33)	4 (18)	ns
	Age (years old)	59 (43–68)	61 (35–69)	54 (45–72)	ns
	Diabetes mellitus, n (%)	4 (17)	3 (25)	10 (46)	ns
	Arterial hypertension, n (%)	15 (63)	6 (50)	14 (74)	ns
	Coronaryischemic disease, n (%)	4 (17)	4 (33)	9 (43)	ns
	Cerebrovasculardisease, n (%)	2 (8)	1 (8)	1 (5)	ns
	Cardiac failure, n (%)	2 (8)	1 (8)	3 (14)	ns
	Peripheralarterialdisease, n (%)	3 (13)	2 (17)	12 (55)	0.004
PD Prescription & Therapeutics	Icodextrin, n (%)	6 (25)	5 (42)	8 (36)	ns
	Use of amino acid solution, n (%)	0 (0)	2 (17)	1 (5)	ns
	Glucose applied (g/day)	120 (114–137)	120 (98–120)	120 (90–120)	ns
	Spironolactone, n (%)	2 (8)	1 (8)	15 (68)	<0.001
	Beta-blockers, n (%)	2 (8)	3 (25)	7 (32)	ns
	Other antihypertensives, n (%)	13 (54)	7 (58)	10 (46)	ns
	Vitamin D analogues, n (%)	15 (63)	10 (83)	16 (73)	ns
	non-calcium Phosphate binders, n (%)	3 (13)	3 (25)	6 (27)	ns
	Vitamin D3 supplements, n (%)	3 (13)	3 (13)	5 (23)	ns
	Cinacalcet, n (%)	6 (25)	3 (25)	6 (27)	ns
	Antiplatelettherapy, n (%)	5 (21)	4 (33)	14 (64)	0.011
	Statins, n (%)	5 (21)	4 (33)	14 (64)	0.011
	Erythropoietin/darbepoetin, n (%)	13 (54)	5 (42)	16 (73)	ns
PD-parameters, Nutrition Status & Frailty	Peritoneal transport (H, HA, L, LA), n (%)	1/14/6/3 (4/58/25/12)	0/7/4/1 (0/58/33/8)	1/11/9/1 (5/50/41/5)	ns
	CA 125 (UI/L)	25.7 (10.8–33.0)	16.5 (11.0–37.3)	16.5 (11.7–23.9)	ns
	nPCR (g/Kg/day)	1.0 (0.76–1.1)	0.89 (0.79–1.1)	0.99 (0.79–1.1)	ns
	Kt/V	2.6 (2.1–3.2)	2.7 (1.8–3.2)	2.7 (2.0–3.3)	ns
	rGFR (mL/min/1.73 m^2^)	6.2 (3.6–9.8)	5.9 (1.1–10.7)	7.9 (5.9–11.2)	ns
	Residual Diuresis (mL)	1700 (1.25–2350)	1350 (280–1650)	1700 (1300–2600)	ns
	Non-Frail, n (%)	18 (75)	9 (75)	15 (68)	ns

Data are presented as median (IQR) or n (%). Biopsy score: S0—no fibrosis, no inflammation, no vasculopathy; S1—no fibrosis, with inflammation and/or vasculopathy; S2—fibrosis; PD—peritoneal dialysis; n—number of individuals; ns—non-significant; H—High, HA—high average, L—low, LA—low average; nPCR—normalized protein catabolic rate; Kt/V—Dialysis efficacy index; rGFR—residual glomerular filtration rate; CA125—cancer antigen 125.

**Table 2 ijms-24-05020-t002:** Baseline variables according to technical failure or MACE.

Study Variable		PD Failure			MACE		
		No (n = 34)	Yes (n = 24)	*p*	No (n = 31)	Yes (n = 27)	*p*
Anthropometric and Clinical data	Women	10 (29)	8 (33)	ns	12 (39)	6 (22)	ns
	Age (years old)	51.0 (37.8–65.5)	65.0 (53.5–71.8)	0.006	51.0 (37.0–63.0)	67.0 (52.0–72.0)	0.003
	Diabetes mellitus	10 (29)	7 (29)	ns	3 (10)	14 (52)	<0.001
	Arterial hypertension	21 (66)	14 (61)	ns	19 (63)	16 (64)	ns
	Coronary ischemic disease	10 (30)	7 (29)	ns	0 (0)	17 (65)	<0.001
	Cerebrovascular disease	3 (9)	1 (4)	ns	0 (0)	4 (15)	0.038
	Cardiac failure	3 (9)	3 (12)	ns	0 (0)	6 (23)	0.006
	Peripheral arterial disease	7 (21)	10 (42)	ns	0 (0)	17 (63)	<0.001
	rGFR (mL/min/1.73 m^2^)	7.9 (5.6–10.2)	6.1 (2.6–10.3)	ns	8.9 (5.3–10.4)	6.0 (2.6–9.3)	ns
	Residual Diuresis (mL)	1700 (1225–2350)	1500 (600–2500)	ns	1700 (1300–2400)	1500 (1000–2350)	ns
PD and Therapeutics Prescription	Icodextrin	14 (41)	5 (21)	ns	10 (32)	9 (33)	ns
	Use of amino acid solution	2 (6)	1 (4)	ns	1 (3)	2 (7)	ns
	Glucose applied	120.0 (90.0–120.0)	120.0 (120.0–143.8)	ns	120.0 (90.0–135.0)	120.0 (90.0–120.0)	ns
	Spironolactone	9 (27)	9 (38)	ns	8 (26)	10 (37)	ns
	Beta–blockers	7 (21)	5 (21)	ns	2 (6)	10 (37)	0.004
	Calcium channels blockers	12 (35)	18 (75)	0.003	16 (52)	14 (52)	ns
	Vitamin D analogues	26 (77)	18 (75)	ns	23 (74)	21 (78)	ns
	Non–calcium Phosphate binders	9 (27)	11 (46)	ns	9 (29)	11 (41)	ns
	Vitamin D3 supplements	24 (71)	17 (71)	ns	20 (64)	21 (78)	ns
	Cinacalcet	8 (24)	7 (29)	ns	9 (29)	6 (22)	ns
	Antiplatelet therapy	10 (30)	8 (33)	ns	0 (0)	18 (69)	<0.001
	Statins	10 (30)	8 (33)	ns	0 (0)	18 (69)	<0.001
	Erythropoietin/darbepoetin	17 (50)	17 (71)	ns	15 (48)	19 (70)	ns
PD-related parameters, Frailty	Peritoneal transport (High)	20 (59)	14 (58)	ns	19 (61)	15 (56)	ns
	CA 125 (UI/L)	14.3 (9.8–25.0)	25.2 (13.5–36.5)	ns	15.6 (10.5–32.3)	16.9 (12.0–36.9)	ns
	nPCR (g/Kg/day)	1.0 (0.81–1.1)	0.87 (0.77–1.1)	ns	1.1 (0.88–1.2)	0.83 (0.78–1.0)	0.023
	Kt/V	2.9 (2.1–3.3)	2.4 (1.9–2.9)	ns	2.9 (2.4–3.4)	2.4 (1.8–3.1)	0.017
	Frail	4 (12)	7 (29)	ns	3 (10)	8 (30)	ns
	Score de Edmonton	2 (2–3)	3.5 (2–6)	0.038	2 (2–3)	4 (2–6)	<0.001
Membrane Fibrosis and Serum biomarkers	α–Klotho (pg/mL)	779 (589–1016)	724 (626–1090)	ns	803 (640–1119)	698 (548–958)	ns
	Galectin–3 (ng/mL)	9.4 (7.50–10.9)	10.4 (9.2–11.0)	0.048	9.9 (8.6–10.8)	10.2 (8.6–11.1)	ns
	FGF21(pg/mL)	1324 (834–2226)	1435 (980–3135)	ns	1410 (967–2061)	1336 (921–2507)	ns
	FGF23 (pg/mL)	748.9 (525.9–862.4)	744.2 (649.2–848.6)	ns	685.9 (578.3–858.8)	761.9 (570.9–850.5)	ns
	TWEAK (pg/mL)	0.14 (0.06–0.49]	0.12 (0.05–0.28)	ns	0.11 (0.06–0.48)	0.16 (0.06–0.27)	ns
	TNF–α (pg/mL)	0.18 (0.13–0.24)	0.16 (0.14–0.20)	ns	0.17 (0.12–0.19)	0.17 (0.15–0.25)	ns
	hs–CRP (μg/mL)	0.35 (0.17–0.48)	0.39 (0.17–0.90)	ns	0.39 (0.16–0.54)	0.35 (0.19–0.63)	ns
	Peritoneal membrane fibrosis	13 (38)	9 (38)	ns	5 (16)	17 (63)	<0.001
	STM	95.0 (40.0–190.0)	60.0 (20.0–200.0)	ns	60.0 (30.0–110.0)	190.0 (50.0–200.0)	0.004
	α–Klotho < 742 pg/mL	16 (47)	13 (54)	ns	12 (39)	17 (63)	ns

Data are present as median (IQR) or n (%). PD—peritoneal dialysis; n—number of individuals; ns—non-significant; nPCR—normalized protein catabolic rate; Kt/V—Dialysis efficacy index; rGFR—residual glomerular filtration rate; CA125—cancer antigen 125; FGF21—Fibroblast growth factor 21; FGF23—Fibroblast growth factor 23 c-terminal; TWEAK—Tumor necrosis factor-like weak inducer of apoptosis; TNFα—Tumor necrosis factor α; hs-CRP—ultra-sensitive C-reactive protein; STM—submesothelial compact zone thickness.

**Table 3 ijms-24-05020-t003:** Baseline variables as predictors of 5-year peritoneal dialysis outcomes.

Study Variables		Time until Event (Month) HR (95% CI), *p* Value	
		PD Failure	MACE
Anthropometric and Clinical data	Women	ns	ns
	Age (years old)	*p* = 0.08	1.044 (1.012–1.077), *p* = 0.04
	Diabetes mellitus	ns	3.717 (1.732–7.978), *p* = 0.01
	Arterial hypertension	ns	ns
	Coronary ischemic disease	ns	10.063 (4.239–23.894), *p* = 0.001
	Cerebrovascular disease	ns	4.206 (1.409–12.582), *p* = 0.27
	Cardiac failure	ns	*p* = 0.059
	Peripheral arterial disease	2.432 (1.066–5.552), *p* = 0.035	8.875 (3.890, 20.248), *p* = 0.001
	GFRr (mL/min/1.73 m^2^)	ns	*p* = 0.110
	Residual Diuresis (mL)	ns	ns
PD and Therapeutics Prescription	Icodextrin	*p* = 0.099	ns
	Use of amino acid solution	ns	ns
	Glucose applied	ns	ns
	Spironolactone	ns	ns
	Beta-blockers	ns	3.518 (1.579–7.838), *p* = 0.001
	Calcium channels blockers	2.88 (1.139–7.236), *p* = 0.017	ns
	Vitamin D analogues	ns	ns
	Non-calcium Phosphate binders	ns	ns
	Vitamin D3 supplements	ns	ns
	Cinacalcet	ns	ns
	Antiplatelet therapy	ns	12.153 (4.982–29.745), *p* = 0.001
	Statins	ns	ns
	Erythropoietin/darbepoetin	ns	ns
PD-related parameters, Frailty	Peritoneal transport (High)	ns	ns
	CA 125 (UI/L)	ns	ns
	nPCR (g/Kg/day)	ns	0.093 (0.011–0.802), *p* = 0.024
	Kt/V	*p* = 0.078	0.475 (0.260–0.869), *p* = 0.013
	Frail	*p* = 0.093	*p* = 0.073
	Score de Edmonton	*p* = 0.150	*p* = 0.051
Membrane Fibrosis and Serum biomarkers	α-Klotho (pg/mL)	ns	*p* = 0.055
	Galectin-3 (ng/mL)	1.271 (0.988–1.635), *p* = 0.042	ns
	FGF21(pg/mL)	*p* = 0.146	ns
	FGF23 (pg/mL)	ns	ns
	TWEAK (pg/mL)	*p* = 0.118	ns
	TNF-α (pg/mL)	*p* = 0.087	ns
	hs-CRP (μg/mL)	ns	ns
	Peritoneal membrane fibrosis	ns	4.181 (1.905–9.175), *p* = 0.001
	STM	ns	1.009 (1.003–1.014), *p* = 0.001
	α-Klotho < 742 pg/mL	ns	*p* = 0.055

Data are present as hazard ratio (95% CI). PD—peritoneal dialysis; n—number of individuals; ns—non-significant; nPCR—normalized protein catabolic rate; Kt/V—Dialysis efficacy index; rGFR—residual glomerular filtration rate; CA125—cancer antigen 125; FGF21—Fibroblast growth factor 21; FGF23—Fibroblast growth factor 23 c-terminal; TWEAK—Tumor necrosis factor-like weak inducer of apoptosis; TNFα—Tumor necrosis factor α; hs-CRP—ultra-sensitive C-reactive protein; STM—submesothelial compact zone thickness.

## Data Availability

Data generated in this study are available from the corresponding author upon reasonable request.

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
