# Peer review of "Fibrosis of Peritoneal Membrane, Molecular Indicators of Aging and Frailty Unveil Vulnerable Patients in Long-Term Peritoneal Dialysis"

_ijms, 2023, doi:10.3390/ijms24055020_

Round 1
Reviewer 1 Report
Extremely interesting study, well presented. I have no specific comment. Minor structural changes at the discretion ofthe editors.
Thank you
Author Response
Reviewer 1
Comment: Extremely interesting study, well presented. I have no specific comment. Minor structural changes at the discretion of the editors.
Response: We are grateful for the appreciation of our work presented by the reviewer. We are convinced that novel evidence is provided that will contribute to our knowledge about uremia and peritoneal dialysis outcomes.
English language and style were checked, improved. All changes and corrections are highlighted in yellow in the revised manuscript
Reviewer 2 Report
Dear Editor and authors
This manuscript is a study which focus on markers of aging or frailty which might indicate for peritoneal dialysis (PD) failure.
The topic and the abstract are interesting. However, the way it presents makes it hard to be understood.
Table 1: the p value "*" might stand for p <0.05 but the explanation is missing. Also the p value is for PD failure (yes/no) comparison or MACE is not clear.(And it should list the EXACT value of each comparison, not a symbol only)
Table 2 and 3 are almost identical and I can not tell the difference. The auhors may condence the message for table 2 and 3 into one table and rewrite the result paragraph for table 2 and 3. The way it present is confusing (whether it is the baseline value or predictor?)
Figure 1 :
The survival curve for galetin 3 is great. why not present TWEAK and FGF23 the same way? Also, the p value of hazard ratio for TWEAK and FGF are significant in Fig 1B, But in Table 3 the p value of comparison for WEAK and FGF 21 are not significant. why?
Figure 2 :
the legend for "D" has two parts. please confirm which one is correct.
D) the same as C) but using Klotho cut-off as a surrogate of membrane fibrosis D) the same as C that consider atherosclerotic arterial disease with the use of antiplatelet therapy.
Minor issure: The word font and arrangement for topic and affliation are different from MDPI setting.
Due to the above mentioned problems, I stop reviewing this manuscript. Please consider carefully rewrite and improve the way its present.
I suggest major revision or reject with resubmission, depends on editor decision.
Author Response
We thank to reviewer the constructive comments. To address the criticisms, we have modified the text accordingly. We are including a revised manuscript file with all changes highlighted in yellow.
Comment: Table 1: the p value "*" might stand for p <0.05 but the explanation is missing. Also the p value is for PD failure (yes/no) comparison or MACE is not clear. (And it should list the EXACT value of each comparison, not a symbol only).
Response: We believe that this comment refers to Table 2. There was a missing column in the submitted Table 2, which justifies the issue questioned by the reviewer. We have replaced the table that now contains the column of the p-value for MACE. Moreover, we substituted the symbol * by the exact p-value to fulfill the referee’s recommendation:
(page 6, results section)
Comment: Table 2 and 3 are almost identical and I can not tell the difference. The auhors may condence the message for table 2 and 3 into one table and rewrite the result paragraph for table 2 and 3. The way it present is confusing (whether it is the baseline value or predictor?)
Response: The referee is completely right. We are sorry for that, the Table 2 and 3 presented at submission were the same. We have corrected it and the table 3 can now be found in page 7, results section
Comment:
Figure 1 :
The survival curve for galetin 3 is great. why not present TWEAK and FGF23 the same way? Also, the p value of hazard ratio for TWEAK and FGF are significant in Fig 1B, But in Table 3 the p value of comparison for WEAK and FGF 21 are not significant. why?
Response: Galectin-3 was the only factor that was related to time until PD failure in univariate analysis (as presented in the correct Table 3). This justifies the choice of galectin 3 for the survival curve (Figure 1A).
Tweak and FGF21 did not reach statistical significance in univariate analysis, but were included in Figure 1B, wherein a multivariate Cox regression model included the variables that showed a p-value < 0.2 and were more relevant in the univariate analysis (Table 3).
Comment: Figure 2 the legend for "D" has two parts. please confirm which one is correct.
- D) the same as C) but using Klotho cut-off as a surrogate of membrane fibrosis D) the same as C that consider atherosclerotic arterial disease with the use of antiplatelet therapy.
Response: Thank you and sorry for that. The legend of the figured was corrected and you can find it now in the manuscript on page 10, results section
“Figure 2. Predictors of time to major cardiovascular event (MACE). (A) Survival curve for time until MACE accordingly to peritoneal membrane status. (B) Multivariate Cox model that includes peritoneal fibrosis as an indicator of the membrane status adjusted for age, nutritional status, frailty and PAD. (C) Multivariate Cox model like B but using α-Klotho cutoff as a surrogate of membrane fibrosis. (D) Multivariate Cox model using levels of circulating α-Klotho as surrogate of membrane fibrosis and using the antiplatelet therapy as an indicator of atherosclerotic arterial disease. (E) Survival curves for time until MACE according to α-Klotho cutoff of 742 pg/mL.
GFR glomerular filtration, PAD peripheral arterial disease, CHD coronary heart disease, HF heart failure, nPCR normalized protein catabolic rate.”
Comment: Minor issure: The word font and arrangement for topic and affliation are different from MDPI setting.
Response: The reviewer is right. We have change it accordingly.
Comment: Due to the above mentioned problems, I stop reviewing this manuscript. Please consider carefully rewrite and improve the way its present. I suggest major revision or reject with resubmission, depends on editor decision
Response: We are sorry for that; we uploaded the wrong version in the moment of the submission. Thank you for your time, patience, and the opportunity. We have changed it accordingly and we believe we have an improved version of the manuscript. English language and style were checked, improved and all changes are highlighted in yellow.
We really appreciate the opportunity to present a revised version of our manuscript that we hope is now suitable for publication at the IJMS.
Reviewer 3 Report
In the results in this manuscript, peritoneal biopsy scores are related to PAD, antiplatelet drugs, statin use, development of MACE and circulating klotho. Klotho is known as a factor associated with arteriosclerosis, and these results suggest that this peritoneal biopsy score simply reflects the state of arteriosclerosis. On the other hand, this peritoneal biopsy score was not associated with peritoneal function or PD failure, and this score is unlikely to have clinical significance. And already, the same authors have reported an association between klotho and this peritoneal biopsy score.
In the present study, the authors show that galectin3 is associated with PD failure. However, galectin3 is known to reflect myocardial injury, arteriosclerosis, and inflammation in various organs including blood vessels. Therefore, it is clear that there are many confounding factors between galectin3 and PD failure. Therefore, the results of this study cannot say anything about the relationship between galectin3 and PD failure, and whether galectin3 is a prognostic factor in PD.
Author Response
Comment: In the results in this manuscript, peritoneal biopsy scores are related to PAD, antiplatelet drugs, statin use, development of MACE and circulating klotho. Klotho is known as a factor associated with arteriosclerosis, and these results suggest that this peritoneal biopsy score simply reflects the state of arteriosclerosis. On the other hand, this peritoneal biopsy score was not associated with peritoneal function or PD failure, and this score is unlikely to have clinical significance. And already, the same authors have reported an association between klotho and this peritoneal biopsy score.
Response: The reviewer is right. These were unexpected findings, but in fact, the presence of fibrosis before the beginning of PD was not related to PD failure or time until failure. Thus, our hypothesis about klotho, as a marker of survival of both membrane and the patient, was proved to be negative.
To better clarify this issue, we added in the discussion section, page 11.
We found that the status of the membrane (evaluated by histomorphology, STM and by a surrogate α-Klotho cutoff) was not associated with changes in the functions of the peritoneal transport. Moreover, the pre-PD membrane status was not predictive of long-term survival of both peritoneal membrane and patients.
As α-Klotho is associated with fibrosis of the peritoneal membrane [4], the absence of association between α-Klotho and PD failure was an unexpected finding. While not considering fibrosis, a previous study about the arteriolar structure concluded that membrane arteriolar frailty in CKD stage 5 patients follows with cardiovascular system damage [34]. Therefore, as α-Klotho is associated with arteriosclerosis and aging, our results might suggest that the peritoneal biopsy score reflects more a vascular vulnerability than the integrity of the membrane. This novel and overlooked dimension might account for the shared mechanisms of persistent uremic phenotype, premature aging, and fibrosis of different tissues. In fact, the membrane might not represent a risk factor but a marker of a particular cardiovascular vulnerability profile.
Comment: In the present study, the authors show that galectin3 is associated with PD failure. However, galectin3 is known to reflect myocardial injury, arteriosclerosis, and inflammation in various organs including blood vessels. Therefore, it is clear that there are many confounding factors between galectin3 and PD failure. Therefore, the results of this study cannot say anything about the relationship between galectin3 and PD failure, and whether galectin3 is a prognostic factor in PD.
Response:
We agree that both uremic molecules (alpha-klotho and galectin3) are associated to myocardial injury, arteriosclerosis, and inflammation. However, differently to klotho, there was a clear association with galectin3 and PD failure. Likewise, unlike klotho, galectin-3 was not associated with membrane fibrosis. Therefore, despite the relation to cardiovascular toxicity, the differences found for both molecules suggest different underlying mechanisms.
To better clarify this issue we have introduced in discussion section:
Differently from α-Klotho, there was a clear association with galectin-3 and PD failure. Galectin-3, which is secreted by macrophages, has been associated with an inflammatory and fibrotic phenotype [19,25–29]. Moreover, Béllon and co-workers (2011) showed that alternative activated macrophages or M2 phenotype were present in the peritoneal effluent drained by patients and were able to stimulate fibroblasts proliferation and the loss of peritoneal function [30].
α-Klotho and galectin-3 share common characteristics eg. both are uremic toxins and have been related to fibrosis and inflammation. However, unlike α-Klotho, galectin-3 was associated with PD failure. Therefore, the differences found in PD outcomes between poor α-Klotho versus rich galectin-3 uremic profiles suggest different underlying mechanisms. Moreover, while low baseline α-Klotho was highly associated with cardiovascular disease, such association was not found for galectin-3 (Table 2). Instead, our data pointed out galectin-3 as a predictor of earlier PD failure. Further studies are necessary to validate this data, but a putative explanation for galectin-3 result is that this molecule is a high affinity binding protein for advanced glycation products [38], which relation to poor membrane efficiency and survival is well accepted [39,40]. Of note, inhibitors of galectin-3 are currently investigated in clinical trials [41–43], although in other areas than PD.
We thank to reviewer the constructive comments. To address the criticisms, we have modified the text accordingly. We are including a revised manuscript file with all changes highlighted in yellow. We really appreciate the opportunity to present a revised version of our manuscript that we hope is now suitable for publication at the IJMS.
Round 2
Reviewer 3 Report
The revised manuscript has been improved in response to the reviewer's comments. This time, there are no particular problems to point out.